# Consistency of Pituitary Adenoma: Prediction by Pharmacokinetic Dynamic Contrast-Enhanced MRI and Comparison with Histologic Collagen Content

**DOI:** 10.3390/cancers13153914

**Published:** 2021-08-03

**Authors:** Kiyohisa Kamimura, Masanori Nakajo, Manisha Bohara, Daigo Nagano, Yoshihiko Fukukura, Shingo Fujio, Tomoko Takajo, Kazuhiro Tabata, Takashi Iwanaga, Hiroshi Imai, Marcel Dominik Nickel, Takashi Yoshiura

**Affiliations:** 1Department of Radiology, Kagoshima University Graduate School of Medical and Dental Sciences, 8-35-1 Sakuragaoka, Kagoshima 890-8544, Japan; masanori@m3.kufm.kagoshima-u.ac.jp (M.N.); k1694685@kadai.jp (M.B.); naga0.0@icloud.com (D.N.); fukukura@m.kufm.kagoshima-u.ac.jp (Y.F.); yoshiura@m3.kufm.kagoshima-u.ac.jp (T.Y.); 2Department of Neurosurgery, Kagoshima University Graduate School of Medical and Dental Sciences, 8-35-1 Sakuragaoka, Kagoshima 890-8544, Japan; ofuji@m2.kufm.kagoshima-u.ac.jp (S.F.); noge5@m2.kufm.kagoshima-u.ac.jp (T.T.); 3Department of Pathology, Kagoshima University Graduate School of Medical and Dental Sciences, 8-35-1 Sakuragaoka, Kagoshima 890-8544, Japan; tabatak@kufm.kagoshima-u.ac.jp; 4Department of Radiological Technology, Kagoshima University Hospital, 8-35-1 Sakuragaoka, Kagoshima 890-8544, Japan; iwanaga@m2.kufm.kagoshima-u.ac.jp; 5MR Research & Collaboration, Siemens Healthcare K.K., 1-11-1 Osaki, Shinagawa, Tokyo 141-8644, Japan; hiroshi.imai@siemens-healthineers.com; 6MR Applications Predevelopment, Siemens Healthcare GmbH, 91052 Erlangen, Germany; marcel.nickel@siemens-healthineers.com

**Keywords:** pituitary adenoma, consistency, magnetic resonance imaging, pharmacokinetic analysis, collagen

## Abstract

**Simple Summary:**

Transsphenoidal resection of hard pituitary adenomas have a particularly high risk of residual tumor and complications. Therefore, prediction of tumor consistency is valuable for planning pituitary adenoma surgery. We prospectively examined whether quantitative pharmacokinetic analysis of dynamic contrast-enhanced magnetic resonance imaging (DCE-MRI) is useful for predicting consistency of pituitary adenoma in 49 participants. We found that the measure of volume of extravascular extracellular space per unit volume of tissue derived from DCE-MRI could predict the consistency of pituitary adenomas. Furthermore, the volume of extravascular extracellular space per unit volume of tissue was significantly positively correlated with histopathologic collagen content of the adenoma. Our results suggest that volume of extravascular extracellular space per unit volume of tissue derived from quantitative pharmacokinetic analysis of DCE-MRI has a predictive value for consistency of pituitary adenomas.

**Abstract:**

Prediction of tumor consistency is valuable for planning transsphenoidal surgery for pituitary adenoma. A prospective study was conducted involving 49 participants with pituitary adenoma to determine whether quantitative pharmacokinetic analysis of dynamic contrast-enhanced magnetic resonance imaging (DCE-MRI) is useful for predicting consistency of adenomas. Pharmacokinetic parameters in the adenomas including volume of extravascular extracellular space (EES) per unit volume of tissue (v_e_), blood plasma volume per unit volume of tissue (v_p_), volume transfer constant between blood plasma and EES (K^trans^), and rate constant between EES and blood plasma (k_ep_) were obtained. The pharmacokinetic parameters and the histologic percentage of collagen content (PCC) were compared between soft and hard adenomas using Mann–Whitney *U* test. Pearson’s correlation coefficient was used to correlate pharmacokinetic parameters with PCC. Hard adenomas showed significantly higher PCC (44.08 ± 15.14% vs. 6.62 ± 3.47%, *p* < 0.01), v_e_ (0.332 ± 0.124% vs. 0.221 ± 0.104%, *p* < 0.01), and K^trans^ (0.775 ± 0.401/min vs. 0.601 ± 0.612/min, *p* = 0.02) than soft adenomas. Moreover, a significant positive correlation was found between v_e_ and PCC (r = 0.601, *p* < 0.01). The v_e_ derived using DCE-MRI may have predictive value for consistency of pituitary adenoma.

## 1. Introduction

Pituitary adenomas are the most common lesions of the sella turcica, comprising approximately 15% of all primary brain tumors [1]. Transsphenoidal surgery is among the standard initial treatments for pituitary adenoma [2]. The main limitation of this technique involves tumor consistency [3]. Most pituitary adenomas with soft consistency can be easily removed using this technique [4,5]. However, tumors with hard consistency may not be successfully debulked via the transsphenoidal approach [6]. Therefore, preoperative information regarding tumor consistency is helpful to neurosurgeons for planning an appropriate surgical approach that avoids or minimizes residual tumor and potential complications [5,6]. Consistency of pituitary adenomas has been considered to be relevant to collagen content [5,6,7,8,9].

There is controversy regarding the value of magnetic resonance imaging (MRI) in predicting consistency of pituitary adenoma. Some studies have shown that relative signal intensity or signal intensity ratio on T1- or T2-weighted MRI and apparent diffusion coefficient (ADC) value may be predictive indicators for tumor consistency, while other have denied their predictive values [2,4,5,6,7,8,9,10,11].

Dynamic contrast-enhanced MRI (DCE-MRI) with pharmacokinetic analysis allow for non-invasive and quantitative assessment of specific tissues [12]. The extended Tofts model is a widely used mathematical model for quantitative pharmacokinetic analysis [13,14,15], and it provides a set of pharmacokinetic parameters including volume of extravascular extracellular space (EES) per unit volume of tissue (v_e_), blood plasma volume per unit volume of tissue (v_p_), volume transfer constant between blood plasma and EES (K^trans^), and rate constant between EES and blood plasma (k_ep_) [16]. DCE-MRI can be used to provide information concerning tumor microvascular distribution, perfusion, and permeability [17,18,19]. In addition, previous studies have reported that the fibrous content in pancreatic cancer is significantly correlated with the values of v_e_ [18]. DCE-MRI of the pituitary gland is performed with a typical temporal resolution of 20 s. Only a few attempts have been made to quantitatively analyze DCE-MRI data of the pituitary gland, which is presumably due to the limited temporal resolution [20]. Recent studies have demonstrated the feasibility of higher temporal resolution DCE-MRI using a prototype compressed sensing volumetric interpolated breath-hold examination (CS VIBE) sequence [21,22].

We hypothesized that v_e_ measurement using the CS VIBE sequence would be useful in predicting the consistency of pituitary adenomas by identifying tumors with increased collagen fibrous content. This study aims to prospectively evaluate values of quantitative pharmacokinetic parameters to preoperatively predict the consistency of pituitary adenomas.

## 2. Materials and Methods

### 2.1. Participant Characteristics

The institutional review board approved this prospective cross-sectional study (approval no. 180255), and informed consent was obtained from each participant.

We included adult (age > 18 years) women and men with pituitary lesions who underwent clinically indicated conventional and DCE-MRI between July 2018 and October 2020 at our institution. Exclusion criteria included pituitary lesions other than adenoma and not undergoing tumorectomy. The hormonal activity of each pituitary adenoma was determined by preoperative measurement of serum hormone levels and pituitary provocation tests. The diagnoses of all the participants were confirmed on the hormonal activity on the bases of guidelines of diagnosis and treatment of hypothalamic pituitary dysfunction [23].

### 2.2. Endocrine Studies

Blood basal levels of the anterior pituitary hormones (growth hormone [GH], thyroid stimulating hormone [TSH], prolactin [PRL], luteinizing hormone [LH], follicle-stimulating hormone [FSH], and adrenocorticotropic hormone [ACTH]) and their target hormones (insulin-like growth factor 1 [IGF-1], free thyroxine [FT4], free triiodothyronine [FT3], testosterone, and estradiol) were measured pre- and postoperatively. Pituitary stimulation tests were also performed in some cases pre- and postoperatively using a combination of thyrotropin-releasing hormone (TRH) (500 μg), LH-releasing hormone (100 μg), and corticotropin-releasing hormone (100 μg). A chemiluminescent enzyme immunoassay was used to measure TSH (normal range 0.50–5.00 μIU/mL). The tumor was considered to be TRH responsive if the TSH level increased more than twice the basal level in response to the TRH stimulation test. Cosecretion of GH was identified by supranormal IGF-1 levels and a lack of GH suppression in response to a 75g glucose tolerance test. Associated hyperprolactinemia was identified when tumor cells showed prolactin immunopositivity. GH hypersecretion was considered to be in complete remission when fulfilling the conditions of a normal basal GH level, normal GH suppression (GH nadir < 0.4 ng/mL) during glucose tolerance testing, and normal IGF-1 levels based on age and sex [24]. Octreotide (50 μg administered subcutaneously) or bromocriptine (2.5 mg by mouth) tests were performed to investigate TSH responses to somatostatin analogs or dopamine agonists. TSH was considered suppressed if it decreased to less than 50% of the basal level. Plasma TSH, FT3, and FT4 levels were measured 2–4 times during the 2-week hospital stay after surgery.

### 2.3. MRI Examinations

All participants underwent MRI with a 3 T scanner (MAGNETOM Prisma; Siemens Healthcare, Erlangen, Germany) and a 20-channel head/neck coil. Our routine imaging for the sellar region included the following pre-contrast sequences (Table 1): coronal pre-contrast 2D T1-weighted spin-echo imaging, coronal 2D T2-weighted turbo spin-echo imaging, and coronal diffusion-weighted imaging using readout-segmented echo-planar, which reduces susceptibility and T2* blurring artifacts [25,26]. For DCE-MRI, coronal T1-weighted imaging was performed using a prototype 3D CS VIBE sequence (Table 1). The DCE-MRI scan was performed after intravenous injection of 0.1 mmol/kg meglumine gadoterate (Magnescope; Guerbet, Aulnay-sous-Bois, France) followed by 20 mL of saline at a rate of 4 mL/s. Subsequently, coronal post-contrast 3D T1-weighted spoiled gradient-echo imaging was performed.

### 2.4. Measurement of Maximum Tumor Diameter and Volume

The maximum tumor diameter was measured by a radiologist (D.N. with 3 years of radiological experience) on coronal post-contrast 3D T1-weighted images using a picture archiving and communication system (Synapse; Fujifilm Medical, Tokyo, Japan). A previously proposed equation (volume = 0.5 × length × height × width) [27] was used to estimate the tumor volume.

### 2.5. Parasellar Extension on MRI (Grading System)

To evalutate radiological characteristics of cavernous sinus involvement, we used a grading system proposed by Knosp et al. [28]. This grading system classifies the parasellar extension of pituitary adenomas on coronal MRI including pre-contrast T1- and T2-weighted images and post-contrast T1-weighted images. Three lines connecting the cross-section of the intracavernous and supracavernous internal carotid arteries distinguish 4 grades of parasellar adenoma extension: a medial tangent, a line through the cross-sectional centers, and a lateral tangent. The adenomas were divided into two groups according to the Knosp grading system: low grade with Knosp grade 0, 1, and 2 tumors and high grade with Knosp grade 3 and 4 tumors, because it was reported that the Knosp grade 0, 1, and 2 tumors demonstrated low rate (0%, 0% and 9.9%) and grade 3 and 4 tumors demonstrated high rate (37.9% and 100%) of cavernous sinus invasion [29].

### 2.6. Pharmacokinetic Analysis of DCE-MRI

Data of DCE-MRI were analyzed using Vitrea (Canon Medical Systems Corporation, Otawara, Japan). The arterial input function was automatically detected at the internal carotid artery. The extended Tofts model was used to calculate tumor pharmacokinetic parameters (v_e_, v_p_, K^trans^, and k_ep_) [16].

The pre-contrast T1-weighted MRIs, T2-weighted MRIs, ADC maps, and pharmacokinetic parameters were automatically co-registered to the post-contrast T1-weighted MRIs in Vitrea. Region of interests (ROIs) were first drawn on the contrast-enhanced MRIs by two independent radiologists (K.K. and M.B. with 23 and 5 years of radiological experience, respectively) who were blinded to participants’ clinical information. These were then duplicated on the other image types. ROIs were manually annotated on the coronal slice with maximum lesion extent. Cystic, necrotic, and hemorrhagic areas were intentionally excluded. An additional round ROI (5 mm diameter) was placed in the normal-appearing white matter of the temporal lobe. The mean signal intensity or parametric value was obtained for each ROI.

For the conventional images, relative signal intensity of the pituitary adenoma was assessed by calculating the ratio of signal intensity on T1-weighted (rT1) and T2-weighted MRI (rT2) in the tumor to those in the normal-appearing white matter.

### 2.7. Intraoperative Findings

Tumor consistency was evaluated during surgery by a neurosurgeon (S.F. with 18 years of neurosurgery experience), who was blinded to the result of quantitative MRI. The tumors were classified into groups of soft and hard consistency: tumors with soft were easily removable through suction, and those with hard consistency were removable with difficulty through suction or not removable through suction but excisable piece by piece [4,5,6,7,8,9,10,11].

### 2.8. Histologic Examination

Histopathological examinations were performed by one pathologist (K.T. with 20 years of experience) who was blinded to the MRI data. Tissues were fixed in 4% paraformaldehyde in 0.05 M phosphate buffer, pH 7.4, followed by immersion in 30% sucrose in 0.05 M phosphate buffer, pH 7.2. Tissues were then processed into cryoblocks with liquid nitrogen and kept in deep freeze. A cryostat was used to obtain 8 μm cryosections, and the tissues were then mounted on glass slides. Tissue conditions were confirmed by hematoxylin–eosin. Azan staining was performed to detect fibrous matrix deposition (Azan staining displays fibrous matrix in blue; other tissue regions are stained in red or purple). Percentage of collagen content (PCC) was obtained using the following methods. Azan-stained histopathological slides were scanned using Aperio CS2 (× 20 magnification; Leica Biosystems, Vista, CA), and images of five randomly chosen areas within the lesions were taken using × 400 magnification. Areas of collagen were measured by automatically tracing collagen content contours using an image processing integration software (WinROOF2015, version 3.12.0; Mitani Corp., Tokyo, Japan). PCC was calculated by using the following equation: PCC = [Σ(Acoll)/Σ(Atum)] × 100, where Acoll is area of collagen and Atum is area of total tumor.

### 2.9. Postoperative MRI Examinations

Each participant in this study received follow-up MRI examinations 6 months postoperatively to determine extent of tumor resection (presence/absence of residual tumor), and yearly follow-up was performed thereafter. Tumor regrowth was positive when a tumor maximum diameter increased by more than 2 mm on MRI from the first follow-up MRI.

### 2.10. Statistical Analyses

Statistical analyses were performed using MedCalc version 15.10.0 (MedCalc Software, Mariakerke, Belgium). Relationships between categorical variables were tested using either the Chi square test or Fisher’s exact test. Comparisons between numerical variables were preformed using either the Mann–Whitney *U* test or unpaired *t*-test. We used the D’Agostino–Pearson normality test to check the normality of the data. Pearson’s correlation coefficients were used to analyze correlations between the MRI parameters, PCC, and hormone levels. Interobserver agreement for MRI parameters was evaluated using intraclass correlation coefficients. Intraclass correlation coefficients of >0.74 were considered excellent agreement [30]. The values measured by the two observers were averaged for each ROI. Receiver operating characteristic curves were generated to calculate the area under the receiver operating characteristic curves, sensitivity, specificity, and accuracy. The maximum Youden index was used to determine the optimal cut-off points. The DeLong method was used to compare area under the receiver operating characteristic curve values [31]. *p* values < 0.05 indicated statistical significance.

## 3. Results

### 3.1. Characteristics of Participants and Adenomas

From 100 participants with pituitary lesions, the following participants were excluded: 26 who had pituitary lesions other than adenoma (histologically confirmed Rathke’s cleft cyst [*n* = 11] and craniopharyngioma [*n* = 6], meningioma [*n* = 4], germinoma [*n* = 2], and arachnoid cyst [*n* = 3] showing typical imaging findings), and 25 who had pituitary adenomas (nonfunctioning [*n* = 17] and PLR producing adenoma [*n* = 8]) without undergoing tumorectomy. A total of 49 participants (mean age 55 ± 17 years; 23 men and 26 women) with pituitary adenoma were included in the final sample (Figure 1). Forty-eight participants were treated with the endoscopic endonasal transsphenoidal technique, and one participant underwent combined transsphenoidal and transcranial resection. Tumor consistency at surgery was classified as soft and hard in 34 (69.4%) and 15 (30.6%) participants, respectively. One participant with TSH producing adenoma treated with somatostatin analogues prior to the operation, and tumor was soft consistency. Three participants had recurrent tumors and were undergoing a second surgery, and all three tumors had hard consistency. No other participants had any prior treatment including radiotherapy. No participant had pituitary apoplexy.

Table 2 shows the characteristics of participants and adenomas. A total of 33 participants were diagnosed as nonfunctioning adenomas, 13 as GH producing adenomas, 2 as TSH producing adenomas, and 1 as an ACTH producing adenoma. There were no significant differences in participants age, gender, hormonal function between soft and hard adenomas.

Hard adenomas had significantly higher Knosp grade than soft adenomas (*p* = 0.01). Hypopituitarism was significantly more frequent in hard adenomas than in soft adenomas (*p* < 0.01). There was no significant difference in the frequency of residual tumor nor tumor regrowth between hard and soft adenomas.

Representative cases are presented in Figure 2 and Figure 3.

### 3.2. Maximum Diameter and Volume of Pituitary Adenomas

The maximum diameter of pituitary adenomas ranged from 6 to 69 mm. Three adenomas (6%) were smaller than 10 mm in maximum diameter (microadenoma), three adenomas (6%) were larger than 40 mm in maximum diameter (giant adenoma), while the remaining 43 (88%) were 10 mm or lager and 40 mm or smaller (macroadenoma). There was no significant difference in maximum diameter between soft and hard adenomas (24.8 ± 14.0 mm vs. 26.1 ± 7.0 mm, *p* = 0.27; Table 2). The volume of pituitary adenomas ranged from 87.5 to 68930 mm^3^. There was no significant difference in volume between soft and hard adenomas (8830 ± 16070 mm^3^ vs. 6420 ± 4210 mm^3^, *p* = 0.18; Table 2).

### 3.3. Interobserver Agreement

The intraclass correlation coefficients and 95% confidence intervals for rT1, rT2, ADC, v_e_, v_p_, K^trans^, and k_ep_ were 0.778 (0.638 to 0.868), 0.940 (0.895 to 0.965), 0.818 (0.699 to 0.893), 0.923 (0.867 to 0.956), 0.980 (0.965 to 0.989), 0.917 (0.857 to 0.952), and 0.882 (0.800 to 0.932), respectively, which indicated excellent agreement for all measures.

### 3.4. Comparisons of Imaging and Histologic Parameters between Nonfunctioning and GH Producing Adenomas

Mean values of the MRI parameters and PCC of the histological examination for nonfunctioning and GH producing adenomas are shown in Table 3. GH producing adenomas had significantly lower rT2 than nonfunctioning adenomas (1.359 ± 0.334 vs. 1.763 ± 0.377, *p* < 0.01). No significant difference was found in any other MRI parameter or PCC.

### 3.5. Correlation of Imaging and Histologic Parameters, and Levels of GH and IGF-1 in Participants with GH-Producing Adenoma

We found a significant positive correlation between v_p_ and level of GH in GH producing adenomas (*n* = 13, r = 0.630, *p* = 0.02). No other MRI parameters and PCC showed a significant correlation with GH (rT1 [r = 0.402, *p* = 0.17], rT2 [r = 0.123, *p* = 0.69], ADC [r = −0.256, *p* = 0.40], v_e_ [r = −0.480, *p* = 0.10], K^trans^ [r = −0.424, *p* = 0.15], k_ep_ [r = −0.333, *p* = 0.27], and PCC [r = −0.273, *p* = 0.37]) or IGF-1 (rT1 [r = 0.294, *p* = 0.33], rT2 [r = −0.019, *p* = 0.95], ADC [r = −0.299, *p* = 0.32], v_e_ [r = −0.497, *p* = 0.08], v_p_ [r = 0.391, *p* = 0.19], K^trans^ [r = −0.397, *p* = 0.18], k_ep_ [r = −0.149, *p* = 0.63], and PCC [r = −0.308, *p* = 0.31]).

### 3.6. Hypopituitarism and Correlation of Imaging and Histologic Parameters, and Cortisol of Nonfunctioning Adenomas

Twenty-one (63.3%) participants with nonfunctioning adenoma had hypopituitarism: 11 (52.4%) participants had cortisol deficits, 3 (14.3%) had FSH deficits, 3 (14.3%) had undergone thyroid agent treatment, and 4 (19.0%) had diabetes insipidus. No MRI parameters or PCC showed a significant correlation with cortisol (rT1 (r = 0.055, *p* = 0.76), rT2 (r = −0.069, *p* = 0.70), ADC (r = −0.036, *p* = 0.84), v_e_ (r = −0.052, *p* = 0.77), v_p_ (r = 0.136, *p* = 0.45), K^trans^ (r = −0.059, *p* = 0.75), k_ep_ (r = −0.073, *p* = 0.69), and PCC (r = −0.118, *p* = 0.51)).

### 3.7. Comparisons of Imaging and Histologic Parameters between the Low and High Grade of Knosp Classification

Mean values of the MRI parameters and PCC of the histological examination for the low and high grade of Knosp classification are shown in Table 4. High grade of Knosp classification adenomas had significantly higher PCC than low grade of Knosp classification adenomas (12.58 ± 15.27 vs. 25.43 ± 22.28, *p* = 0.03). No significant difference was found in any other MRI parameter.

### 3.8. Comparisons of Imaging and Histologic Parameters between Total Resection and Residual Tumor

Mean values of the MRI parameters and PCC of the histological examination for total resection and residual tumor are shown in Table 5. There were no significant differences in any parameter.

### 3.9. Comparisons of Imaging and Histologic Parameters between Soft and Hard Adenomas of All Adenomas

Mean values of the MRI parameters and PCC of the histological examination for soft and hard adenomas are shown in Table 6. Hard adenomas had significantly higher PCC than soft adenomas (44.08 ± 15.14 vs. 6.62 ± 3.47, *p* < 0.01). Hard adenomas had significantly higher v_e_ (0.332 ± 0.124 vs. 0.221 ± 0.104, *p* < 0.01) and K^trans^ (0.775 ± 0.401/min vs. 0.601 ± 0.612/min, *p* = 0.02) values than soft adenoma, whereas there were no significant differences in rT1 (0.854 ± 0.088 vs. 0.840 ± 0.077, *p* = 0.40), rT2 (1.580 ± 0.295 vs. 1.661 ± 0.453, *p* = 0.55), ADC (0.769 ± 0.201 × 10^−3^ mm^2^/s vs. 0.832 ± 0.263 × 10^−3^ mm^2^/s, *p* = 0.47), v_p_ (0.040 ± 0.042 vs. 0.070 ± 0.084, *p* = 0.36), or k_ep_ (2.641 ± 1.672/min vs. 2.240 ± 1.691/min, *p* = 0.37) values (Table 6).

### 3.10. Comparisons of Imaging and Histologic Parameters between Soft and Hard Nonfunctioning Adenomas

Mean values of the MRI parameters and PCC of the histological examination for soft and hard nonfunctioning adenomas are shown in Table 7. Hard nonfunctioning adenomas had significantly higher PCC (43.08 ± 15.91 vs. 6.62 ± 3.51, *p* < 0.01) and v_e_ (0.310 ± 0.114 vs. 0.215 ± 0.118, *p* = 0.03), and significantly lower v_p_ (0.036 ± 0.043 vs. 0.089 ± 0.096, *p* = 0.04) and rT2 (1.558 ± 0.273 vs. 1.880 ± 0.383, *p* = 0.01) values than soft nonfunctioning adenoma, whereas there were no significant differences in the other MRI parameters (Table 7).

### 3.11. MRI Parameters Correlated with Percentage of Collagen Content in Pituitary Adenomas

We found a significant positive correlation between v_e_ and PCC in pituitary adenomas (*n* = 49, r = 0.601, *p* < 0.01; Figure 4). No other MRI parameters showed a significant correlation with PCC (rT1 [r = 0.035, *p* = 0.81], rT2 [r = −0.075, *p* = 0.61], ADC [r = 0.044, *p* = 0.76], v_p_ [r = −0.165, *p* = 0.26], K^trans^ [r = 0.057, *p* = 0.70], and k_ep_ [r = −0.055, *p* = 0.71]).

In addition, we found a significant positive correlation between v_e_ and PCC in nonfunctioning adenomas (*n* = 33, r = 0.567, *p* < 0.01), whereas no significant correlation with PCC was found in the other MRI parameters (rT1 [r = 0.157, *p* = 0.38], rT2 [r = −0.309, *p* = 0.08], ADC [r = −0.063, *p* = 0.73], v_p_ [r = −0.238, *p* = 0.18], K^trans^ [r = 0.033, *p* = 0.85], and k_ep_ [r = −0.045, *p* = 0.81]).

### 3.12. Diagnostic Performance

Figure 5 shows the receiver operating characteristic curves for the MRI parameters. The area under the receiver operating characteristic curves for rT1, rT2, ADC, v_e_, v_p_, K^trans^, and k_ep_ were 0.578, 0.555, 0.566, 0.712, 0.583, 0.703, and 0.582, respectively (Table 8). Among the MRI parameters, v_e_ was the most discriminative quantitative measurement for predicting hard adenomas; however, the area under the receiver operating characteristic curve for v_e_ did not significantly differ from other parameters. Table 4 shows sensitivity, specificity, and accuracy at the optimal cut-off values of each MRI parameter for the diagnosis of hard adenomas. Among all MRI parameters, v_e_ showed the highest accuracy (81.6%) among all MRI parameters.

## 4. Discussion

The present study suggests that preoperative mapping of volume of extravascular extracellular space per unit volume of tissue derived from pharmacokinetic dynamic contrast-enhanced MRI is a useful imaging method for predicting the consistency of pituitary adenomas. Tumor consistency can critically affect surgical resectability of pituitary adenomas in participants who have been treated using a transsphenoidal endoscopic technique [5,6]. In most cases, pituitary adenoma has a soft consistency, and thus, aspiration and curettage are typically used to remove the tumor. However, 5–15% of large pituitary adenomas have a hard consistency [9], and these tumors require resection using a surgical knife [32], which may increase the risk of complications, such as damage of the healthy pituitary gland, internal carotid artery, and optic nerve. Moreover, transsphenoidal resection of hard adenomas can result in residual tumor tissue and may require a second-look surgery, radiotherapy, or additional transcranial surgery. Therefore, imaging prediction of tumor consistency is valuable for planning transsphenoidal surgery for pituitary adenoma. To the best of our knowledge, this is the first study to reveal the utility of quantitative pharmacokinetic analysis of dynamic contrast-enhanced MRI in predicting the consistency of pituitary adenoma.

Consistency of pituitary adenoma has been considered to be relevant to collagen content [5,8,9,32]. On the basis of Azan, van Gieson, and Sirius red staining methods, previous studies have reported mean percentages of collagen contents of hard adenomas were significantly higher than those of soft adenomas [9,33,34,35]. This study confirmed a relative abundancy of collagen, as measured by histopathological PCC, in hard adenomas compared with soft adenomas. In pancreatic cancer, extracellular fibrosis content has been shown to be positively correlated to v_e_ obtained using pharmacokinetic analysis of DCE-MRI [18]. Collagen content is also one of the extravascular extracellular matrices in pituitary adenomas; thus, it is conceivable that higher PCC is associated with an increase in v_e_. As expected, this study revealed that v_e_ was positively correlated with PCC in pituitary adenomas.

Despite the routine implementation of DCE-MRI as a preoperative MRI protocol for pituitary adenoma, quantitative pharmacokinetic analysis of pituitary DCE-MRI has rarely been reported. Zhai et al. [20] and Liu et al. [36] performed pharmacokinetic analysis of pituitary DCE-MRI using T1-weighted gradient-echo pulse sequences with temporal resolutions of 8 to 8.5 s and an in-plane resolution of <1 mm^2^ but comparatively thicker slices ranging from 2.5 to 3 mm. In this study, we employed the compressed sensing incorporated in a 3D gradient-echo sequence to enable both higher temporal resolution (5 s) and submillimeter isotropic voxels (0.9 × 0.9 × 0.9 mm^3^) [37]. This type of imaging technique allows quantitative pharmacokinetic DCE-MRI of smaller pituitary lesions.

In addition to the pharmacokinetic parameters, we tested the possible utility of conventional MRI parameters for the prediction of pituitary adenoma consistency. Previous studies have reported conflicting results. Ma et al. [8] showed that relative signal intensity (tumor to frontal lobe white matter) on pre-contrast T1-weighted images are useful in predicting the consistency of pituitary adenoma, whereas other studies have suggested that there is no relationship between tumor consistency and signal intensity on T1-weighted MRIs [2,3,4,5,6]. Furthermore, another study described an inverse correlation between signal intensity on T2-weighted MRIs and collagen content, which was abundant in hard pituitary adenomas [9]. However, several studies have failed to show such a relationship [2,3,4,5,6,7]. One study demonstrated that lower ADC values are correlated with softer tumor consistency at surgery and with higher cellularity at pathology [5]. In contrast, there has also been a report that showed that lower ADC values are correlated with harder tumor consistency and higher collagen content [7]. We did not find significant correlations between rT1, rT2, and ADC maps with tumor consistency or collagen content. Our study showed that the hard adenomas have significantly higher Knosp score and residual tumor than soft adenomas. Our results are in line to previous literature, which showed tumors extending into the cavernous sinuses had a higher consistency grade [38]. In our study, none of rT1, rT2, or ADC was significantly correlated with tumor consistency or collagen content when all adenoma subtypes were included in the analysis. However, the present study revealed that rT2 of hard nonfunctioning adenoma was significantly lower than that of soft nonfunctioning adenoma, while no significant difference was noted in rT1 nor ADC. It was suggested that in addition to v_e_, rT2 may be useful in predicting consistency of nonfunctioning adenomas.

Previous studies have implicated GH in the progression of several cancers, including breast, colorectal, and pancreatic [39,40,41]. A mechanism by which GH may play this role in cancer is through the induction of the epithelial-to-mesenchymal transition [42] and may lead theoretically to increase in fibroblasts [42,43]. However, our study failed to show a significant correlation between the GH level and PCC. The GH level showed a significant positive correlation with v_p_, which is blood plasma volume per unit volume of tissue. The underlying mechanism of this correlation is unknown.

Our study showed that the rT2 of GH producing adenomas was significantly lower than that of nonfunctioning adenomas, which is compatible with literature [34,44]. It was previously reported that T2 signal intensity is correlated with granulation pattern, collagen content, degree of fibrosis, and amyloid accumulation in GH producing adenomas [34,44].

There were two TSH producing adenomas in our series, and both had soft consistency. It was previously reported that TSH producing adenomas are characterized by fibrosis and hard tumor consistency [45,46], which was not the case in our series. Future studies with larger number of participants are required to clarify this point.

In our series, only one participant received preoperative treatment with somatostatin analogues, which may possibly influence microenvironment in adenoma by altering hormonal secretion, cytokine, and growth factor secretome [47]. A previous study reported that there was no significant change in T2 signal intensity ratio between before and after somatostatin analogue treatment in GH producing adenomas [48]. Effects of somatostatin analogue treatment on MRI parameters and tumor consistency need to be clarified in future studies.

In our study, the cortisol level did not influence MRI parameters or PCC in participants with nonfunctioning adenomas. This result did not support a hypothesis that participants with low cortisol may have decreased blood pressure and circulation, which might be affecting the dynamic MRI parameters.

No imaging parameter was significantly different between totally resected tumors and those with residual tumor, and we failed to show that v_e_ is a direct predictor for residual tumor. However, the mean value of v_e_ of adenomas with residual tumor was higher than that of totally resected tumors. Therefore, it would be worth further studies as a candidate for predictor of surgical outcomes. Tumor regrowth was seen in only 4 cases, therefore we could not statistically compare MRI parameters and PCC between tumors with and without regrowth. Future studies with larger number of participants are required to clarify the roles of MRI parameters in predicting outcomes of pituitary adenoma.

This study has some limitations. First is the use of a small sample size. Further studies should study a larger sample of participants to validate the potential threshold value of v_e_ for tumor resectability using aspiration. Second, we did not perform immunohistochemistry analyses for pituitary transcription factors to define the lineage of the pituitary tumors according to WHO 2017 classification. Instead, each participant was diagnosed according to the guideline of diagnosis and treatment of hypothalamic pituitary dysfunction published by The Japan Endocrine Society in 2019 [23]. Correlations between the full classification of the pituitary adenoma and MRI parameters remain to be a subject of future studies. Finally, assessment of tumor consistency was carried out qualitatively by a single operating surgeon, so possible observer bias cannot be excluded. Nonetheless, similar qualitative intraoperative assessments of pituitary tumor consistency have been used in many previous studies [4,5,6,7,8,9,10,11]. Moreover, the relationship between v_e_ and tumor consistency was supported by the significant positive correlation between v_e_ and histological collagen content.

## 5. Conclusions

We performed quantitative pharmacokinetic analysis of pituitary adenomas using compressed sensing-based high-temporal resolution dynamic contrast-enhanced MRI. Our results suggest that volume of extravascular extracellular space per unit volume of tissue derived from quantitative pharmacokinetic analysis provides valuable information regarding the consistency of pituitary adenomas.

## Figures and Tables

**Figure 1 cancers-13-03914-f001:**
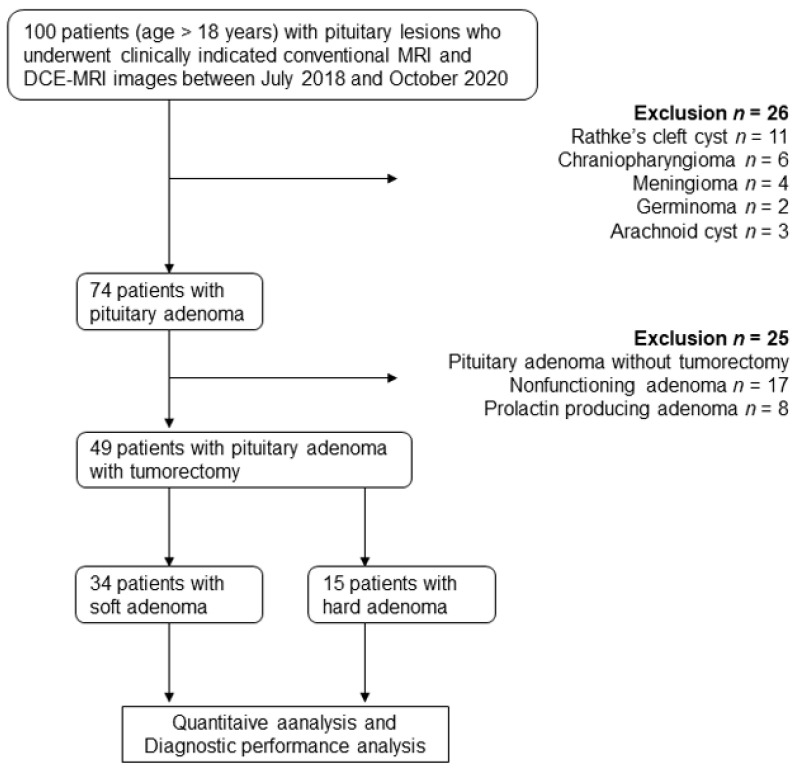
Study flowchart for the inclusion and exclusion criteria of the participant sample and pituitary lesion characterization. DCE-MRI: dynamic contrast-enhanced MRI.

**Figure 2 cancers-13-03914-f002:**
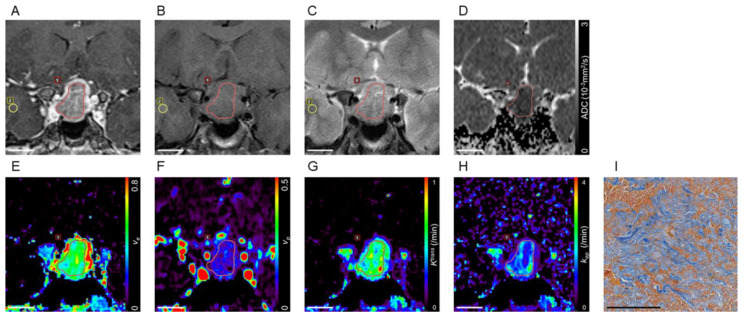
Pituitary adenomas with hard consistency. A contrast-enhanced coronal T1-weighted image of a pituitary adenoma extending into the suprasellar cistern (**A**). The adenoma is slightly hypointense in the white matter on the T1-weighted image (rT1 = 0.931; **B**) and hyperintense on the T2-weighted image (rT2 = 1.71; **C**). The corresponding ADC map shows a low ADC value (0.674 × 10^−3^mm^2^/s; **D**). Pharmacokinetic parametric mapping shows a high v_e_ value (0.440; **E**), a low v_p_ value (0.050; **F**), a high K^trans^ value (0.615/min; **G**), and a low k_ep_ value (1.410/min; **H**) in the adenoma. Region of interests for tumor and normal-appearing white matter are indicated by red and yellow lines, respectively. Azan-stained section of tumor at pathologic examination shows abundant collagen in the stromal fibrous tissue in blue and small tumor cells in red (PCC = 50.9%; **I**). Bar indicates 20 mm (**A**–**H**), 200 μm (**I**). rT1: ratio of signal intensity on T1-weighted MRI; rT2: ratio of signal intensity on T2-weighted MRI; ADC: apparent diffusion coefficient; v_e_: volume of extravascular extracellular space per unit volume of tissue; v_p_: blood plasma volume per unit volume of tissue; K^trans^: volume transfer constant between blood plasma and extravascular extracellular space; k_ep_: rate constant between extravascular extracellular space and blood plasma; PCC: percentage of collagen content.

**Figure 3 cancers-13-03914-f003:**
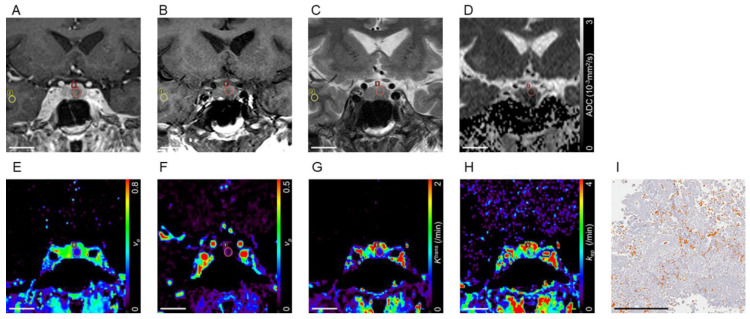
Pituitary adenoma with soft consistency. A contrast-enhanced coronal T1-weighted image of a pituitary microadenoma which maximum diameter is 6 mm (**A**). The adenoma is hypointense in the white matter on the T1-weighted image (rT1 = 0.796; **B**) and hyperintense on the T2-weighted image (rT2 = 1.422; **C**). The corresponding ADC map shows a low ADC value (0.581 × 10^−3^mm^2^/s; **D**). Parametric mapping shows a low v_e_ value (0.205; **E**), a low v_p_ value (0.030; **F**), a low K^trans^ value (0.395/min; **G**), and a high k_ep_ value (1.900/min; **H**) in the adenoma. Region of interests for tumor and normal-appearing white matter are indicated by red and yellow lines, respectively. Azan-stained section of tumor at pathologic examination shows scant collagen in the stromal fibrous tissue in blue within predominant tumor cells in red (PCC = 6.1%; **I**). Bar indicates 20 mm (**A**–**H**), 200 μm (**I**). rT1: ratio of signal intensity on T1-weighted MRI; rT2: ratio of signal intensity on T2-weighted MRI; ADC: apparent diffusion coefficient; v_e_: volume of extravascular extracellular space per unit volume of tissue; v_p_: blood plasma volume per unit volume of tissue; K^trans^: volume transfer constant between blood plasma and extravascular extracellular space; k_ep_: rate constant between extravascular extracellular space and blood plasma; PCC: percentage of collagen content.

**Figure 4 cancers-13-03914-f004:**
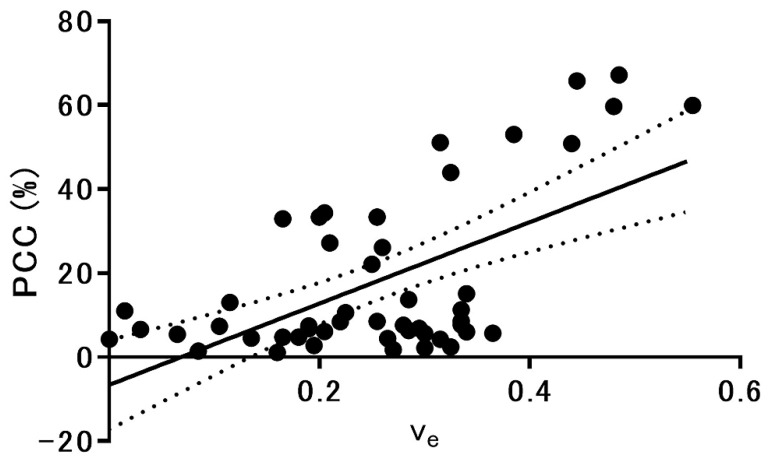
Scatterplot shows the relationship between v_e_ and the PCC in 49 pituitary adenomas. The solid and dotted lines indicate the linear regression line and 95% confidence intervals. v_e_: volume of extravascular extracellular space per unit volume of tissue; PCC: percentage of collagen content.

**Figure 5 cancers-13-03914-f005:**
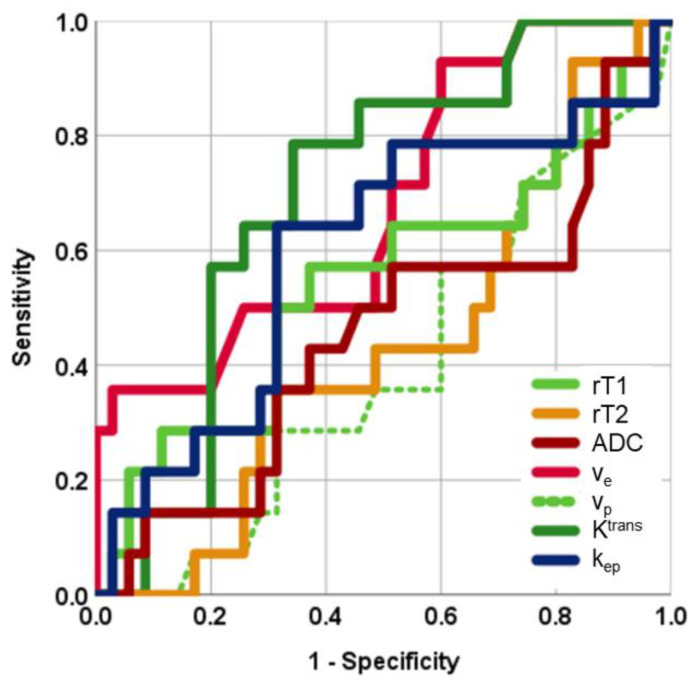
Receiver operating characteristic curve of each parameter for differentiating hard adenomas from soft adenomas (34 soft adenomas and 15 hard adenomas). rT1: ratio of signal intensity on T1-weighted MRI; rT2: ratio of signal intensity on T2-weighted MRI; ADC: apparent diffusion coefficient; v_e_: volume of extravascular extracellular space per unit volume of tissue; v_p_: blood plasma volume per unit volume of tissue; K^trans^: volume transfer constant between blood plasma and extravascular extracellular space; k_ep_: rate constant between extravascular extracellular space and blood plasma.

**Table 1 cancers-13-03914-t001:** Pituitary MRI protocol.

MRI Protocol	Pre-Contrast T1-Weighted Imaging	T2-Weighted Imaging	DWI	DCE Imaging	Post-Contrast T1-Weighted Imaging
Orientation	Coronal	Coronal	Coronal	Coronal	Coronal
Sequence	2D SE	2D TSE	RESOLVE	3D VIBE	3D FLASH
TR (ms)	450	4000	4000	3.9	4.68
TE (ms)	12	95	49	0.97	1.74
FA (degree)	70/180	90/130	150	10	11
Bandwidth (Hz/pixel)	130	189	694	670	370
Number of excitations	1	2	1 and 3 for low and high b values	1	2
Turbo factor	N/A	9	N/A	N/A	N/A
Acceleration factor	1	2	2	N/A	N/A
CS factor	N/A	N/A	N/A	7	N/A
b values (s/mm^2^)	N/A	N/A	0 and 1000	N/A	N/A
Directions of motion-probing gradients	N/A	N/A	3	N/A	N/A
Readout segments	N/A	N/A	9	N/A	N/A
FOV (mm^2^)	180 × 180	180 × 180	120 × 120	159 × 180	200 × 200
Matrix	240 × 320	313 × 448	84 × 120	163 × 192	320 × 256
Thickness (mm)	3	3	2	0.9	0.6
Intersection gap (mm)	0.3	0.3	0	N/A	N/A
Temporal resolution (s)	N/A	N/A	N/A	5	N/A
Acquisition time (s)	148	158	252	203	161

DWI: diffusion weighted image; DCE: dynamic contrast-enhanced; TR: repetition time; TE: echo time; FA: flip angle; CS: compressed sensing; FOV: field of view; SE: spin-echo; TSE: turbo spin-echo; RESOLVE: readout-segmented echo-planar; VIBE: volumetric interpolated breath-hold examination; FLASH: fast low-angle shot; N/A: not applicable.

**Table 2 cancers-13-03914-t002:** Characteristics of participants and adenomas.

Participants Characteristics	Total (*n* = 49)	Soft Adenoma (*n* = 34)	Hard Adenoma (*n* = 15)	*p* Value
Age (y)	55 ± 17	53 ± 17	60 ± 17	0.12 ^a^
No. of men	23 (46.9)	15 (44.1)	8 (53.3)	0.76 ^a^
Maximum diameter (mm)	24.2 ± 12.2	24.8 ± 14.0	26.1 ± 7.0	0.27 ^a^
Volume (mm^3^)	8090 ± 13,570	8830 ± 16,070	6420 ± 4210	0.18 ^a^
Pituitary lesions				
Nonfunctioning	33 (67.3)	21 (61.8)	12 (80.0)	0.32 ^b^
Functioning	16 (32.7)	13 (38.2)	3 (20.0)	
GH producing	13 (26.5)	10 (29.4) Densely granulated	3 (20.0) Sparsely granulated	
TSH producing	2 (4.1)	2 (5.9) Densely granulated	0 (0)	
ACTH producing	1 (2.1)	1 (2.9) Densely granulated	0 (0)	
Knosp classification				
0	3 (6.1)	3 (8.8)	0 (0)	0.01 ^a^
1	13 (26.5)	11 (32.4)	2 (13.3)	
2	12 (24.5)	10 (29.4)	2 (13.3)	
3	18 (36.8)	8 (23.5)	10 (66.7)	
4	3 (6.1)	2 (5.9)	1 (6.7)	
Residual tumor	12 (24.5)	6 (17.6)	6 (40.0)	0.09 ^c^
Hypopituitarism	24 (49.0)	13 (38.2)	11 (73.3)	<0.01 ^c^
Tumor regrowth	4 (8.2)	3 (8.8)	1 (6.7)	

GH: growth hormone; TSH: thyrotropin-releasing hormone; ACTH: adrenocorticotropic hormone. Statistical tests used: ^a^ Mann–Whitney *U* test, ^b^ Fisher’s exact test, ^c^ Chi squared test.

**Table 3 cancers-13-03914-t003:** MRI parameters and histologic percentage of collagen content of nonfunctioning and GH producing adenomas.

Parameters	Nonfunctioning (*n* = 33)	GH-Producing (*n* = 13)	*p* Value
rT1	0.837 ± 0.079	0.850 ± 0.074	0.81 ^a^
rT2	1.763 ± 0.377	1.359 ± 0.335	<0.01 ^a^
ADC (10^−3^ mm^2^/s)	0.856 ± 0.270	0.748 ± 0.121	0.18 ^a^
v_e_	0.250 ± 0.124	0.266 ± 0.131	0.70 ^b^
v_p_	0.069 ± 0.084	0.047 ± 0.052	0.70 ^a^
K^trans^ (/min)	0.734 ± 0.646	0.482 ± 0.301	0.40 ^a^
k_ep_ (/min)	2.558 ± 1.891	1.918 ± 1.147	0.26 ^b^
PCC (%)	19.88 ± 20.33	15.61 ± 19.51	0.35 ^a^

GH: growth hormone; rT1: ratio of signal intensity on T1-weighted MRI; rT2: ratio of signal intensity on T2-weighted MRI; ADC: apparent diffusion coefficient; v_e_: volume of extravascular extracellular space per unit volume of tissue; v_p_: blood plasma volume per unit volume of tissue; K^trans^: volume transfer constant between blood plasma and extravascular extracellular space; k_ep_: rate constant between extravascular extracellular space and blood plasma; PCC: percentage of collagen content. Statistical tests used: ^a^ Mann–Whitney *U* test, ^b^ unpaird *t*-test.

**Table 4 cancers-13-03914-t004:** MRI parameters and histologic percentage of collagen content of the low and high grade of Knosp classification.

Parameters	Low Grade (*n* = 28)	High Grade (*n* = 21)	*p* Value
rT1	0.842 ± 0.081	0.847 ± 0.080	0.85 ^a^
rT2	1.590 ± 0.467	1.697 ± 0.319	0.37 ^a^
ADC (10^−3^ mm^2^/s)	0.849 ± 0.292	0.764 ± 0.156	0.20 ^b^
v_e_	0.237 ± 0.116	0.279 ± 0.126	0.24 ^a^
v_p_	0.065 ± 0.086	0.056 ± 0.057	0.69 ^b^
K^trans^ (/min)	0.617 ± 0.534	0.703 ± 0.608	0.61 ^b^
k_ep_ (/min)	2.268 ± 1.629	2.490 ± 1.773	0.66 ^b^
PCC (%)	12.58 ± 15.27	25.43 ± 22.28	0.03 ^b^

rT1: ratio of signal intensity on T1-weighted MRI; rT2: ratio of signal intensity on T2-weighted MRI; ADC: apparent diffusion coefficient; v_e_: volume of extravascular extracellular space per unit volume of tissue; v_p_: blood plasma volume per unit volume of tissue; K^trans^: volume transfer constant between blood plasma and extravascular extracellular space; k_ep_: rate constant between extravascular extracellular space and blood plasma; PCC: percentage of collagen content. Statistical tests used: ^a^ unpaird *t*-test, ^b^ Mann–Whitney *U* test.

**Table 5 cancers-13-03914-t005:** MRI parameters and histologic percentage of collagen content of total resection and residual tumor.

Parameters	Total Resection (*n* = 36)	Residual Tumor (*n* = 13)	*p* Value
rT1	0.854 ± 0.080	0.819 ± 0.078	0.18 ^a^
rT2	1.650 ± 0.444	1.600 ± 0.310	0.71 ^a^
ADC (10^−3^ mm^2^/s)	0.820 ± 0.267	0.793 ± 0.179	0.69 ^b^
v_e_	0.243 ± 0.121	0.288 ± 0.120	0.25 ^a^
v_p_	0.062 ± 0.075	0.057 ± 0.077	0.85 ^b^
K^trans^ (/min)	0.567 ± 0.466	0.894 ± 0.739	0.16 ^b^
k_ep_ (/min)	2.109 ± 1.472	3.067 ± 2.052	0.14 ^b^
PCC (%)	16.45 ± 18.71	22.64 ± 21.58	0.37 ^b^

rT1: ratio of signal intensity on T1-weighted MRI; rT2: ratio of signal intensity on T2-weighted MRI; ADC: apparent diffusion coefficient; v_e_: volume of extravascular extracellular space per unit volume of tissue; v_p_: blood plasma volume per unit volume of tissue; K^trans^: volume transfer constant between blood plasma and extravascular extracellular space; k_ep_: rate constant between extravascular extracellular space and blood plasma; PCC: percentage of collagen content. Statistical tests used: ^a^ Mann–Whitney *U* test, ^b^ unpaird *t*-test.

**Table 6 cancers-13-03914-t006:** MRI parameters and histologic percentage of collagen content of soft and hard adenomas of all adenomas.

Parameters	Tumor Consistency Group		*p* Value
	Soft Adenoma (*n* = 34)	Hard Adenoma (*n* = 15)	
rT1	0.840 ± 0.077	0.854 ± 0.088	0.40 ^a^
rT2	1.661 ± 0.453	1.580 ± 0.295	0.55 ^a^
ADC (10^−3^ mm^2^/s)	0.832 ± 0.263	0.769 ± 0.201	0.47 ^a^
v_e_	0.221 ± 0.104	0.332 ± 0.124	<0.01 ^b^
v_p_	0.070 ± 0.084	0.040 ± 0.042	0.36 ^a^
K^trans^ (/min)	0.601 ± 0.612	0.775 ± 0.401	0.02 ^a^
k_ep_ (/min)	2.240 ± 1.691	2.641 ± 1.672	0.37 ^a^
PCC (%)	6.62 ± 3.47	44.08 ± 15.14	<0.01 ^b^

rT1: ratio of signal intensity on T1-weighted MRI; rT2: ratio of signal intensity on T2-weighted MRI; ADC: apparent diffusion coefficient; v_e_: volume of extravascular extracellular space per unit volume of tissue; v_p_: blood plasma volume per unit volume of tissue; K^trans^: volume transfer constant between blood plasma and extravascular extracellular space; k_ep_: rate constant between extravascular extracellular space and blood plasma; PCC: percentage of collagen content. Statistical tests used: ^a^ Mann–Whitney *U* test, ^b^ unpaird *t*-test.

**Table 7 cancers-13-03914-t007:** MRI parameters and histologic percentage of collagen content of soft and hard nonfunctioning adenomas.

Parameters	Tumor Consistency Group		*p* Value
	Soft Adenoma (*n* = 21)	Hard Adenoma (*n* = 12)	
rT1	0.824 ± 0.071	0.861 ± 0.090	0.24 ^a^
rT2	1.880 ± 0.383	1.558 ± 0.273	0.01 ^a^
ADC (10^−3^ mm^2^/s)	0.913 ± 0.285	0.756 ± 0.217	0.09 ^a^
v_e_	0.215 ± 0.118	0.310 ± 0.114	0.03 ^b^
v_p_	0.089 ± 0.096	0.036 ± 0.043	0.04 ^a^
K^trans^ (/min)	0.680 ± 0.752	0.829 ± 0.413	0.47 ^a^
k_ep_ (/min)	2.365 ± 2.006	2.894 ± 1.701	0.43 ^a^
PCC (%)	6.62 ± 3.51	43.08 ± 15.91	<0.01 ^a^

rT1: ratio of signal intensity on T1-weighted MRI; rT2: ratio of signal intensity on T2-weighted MRI; ADC: apparent diffusion coefficient; v_e_: volume of extravascular extracellular space per unit volume of tissue; v_p_: blood plasma volume per unit volume of tissue; K^trans^: volume transfer constant between blood plasma and extravascular extracellular space; k_ep_: rate constant between extravascular extracellular space and blood plasma; PCC: percentage of collagen content. Statistical tests used: ^a^ Mann–Whitney *U* test, ^b^ unpaird *t*-test.

**Table 8 cancers-13-03914-t008:** Diagnostic performance of MRI parameters in detecting hard adenomas.

Parameters	AUC	Optimal Cut-Off Value	Sensitivity (%)	Specificity (%)	Accuracy (%)
rT1	0.578	0.866	60.0 (9/15)	64.7 (22/34)	63.3 (31/49)
rT2	0.555	1.551	53.3 (8/15)	64.7 (22/34)	61.2 (30/49)
ADC	0.566	0.674 (10^−3^ mm^2^/s)	46.7 (7/15)	73.5 (25/34)	65.3 (32/49)
v_e_	0.712	0.365	40.0 (6/15)	100 (34/34)	81.6 (40/49)
v_p_	0.583	0.100	93.3 (14/15)	26.5 (9/34)	46.9 (23/49)
K^trans^	0.703	0.560 (/min)	80.0 (12/15)	67.7 (23/34)	71.4 (35/49)
k_ep_	0.582	1.980 (/min)	60.0 (9/15)	67.7 (23/34)	65.3 (40/49)

rT1: ratio of signal intensity on T1-weighted MRI; rT2: ratio of signal intensity on T2-weighted MRI; v_e_: volume of extravascular extracellular space per unit volume of tissue; v_p_: blood plasma volume per unit volume of tissue; K^trans^: volume transfer constant between blood plasma and extravascular extracellular space; k_ep_: rate constant between extravascular extracellular space and blood plasma.

## Data Availability

The data presented in this study are available on request from the corresponding authors. The data are not publicly available due to restrictions of the institutional Ethics Committee statement.

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
