# Peer review of "Consistency of Pituitary Adenoma: Prediction by Pharmacokinetic Dynamic Contrast-Enhanced MRI and Comparison with Histologic Collagen Content"

_cancers, 2021, doi:10.3390/cancers13153914_

Round 1
Reviewer 1 Report
Dear Authors,
I think the authors answered each and every one of our questions and comments.
This revised manuscript is acceptable.
Thank you!
Reviewer 2 Report
The authors addressed all the comments and points I've previously raised in a satisfactory manner. I feel that the quality and relevance of the manuscript has significantly improved. No further comments or questions from my end.
Reviewer 3 Report
The authors have made a concerted effort at responding to the questions and comments raised earlier.
Reviewer 4 Report
Thank you for your careful responses to my questions. I have no other concerns at this time.
This manuscript is a resubmission of an earlier submission. The following is a list of the peer review reports and author responses from that submission.
Round 1
Reviewer 1 Report
In this manuscript, the authors investigated whether quantitative pharmacokinetic analysis of DCE-MRI is associated with consistency of pituitary adenomas. They showed higher volume of extravascular extracellular space (EES) per unit volume of tissue (ve) is more in hard adenomas, and the ve is associated with the histologic percentage of collagen con- 42
tent (PCC).
This analysis is very unique, and prediction of hardness is clinically useful.
However, there are several concerns.
Comments;
- Definition of hard and soft of pituitary adenomas is ambiguous. If hardness is the main outcome, it should be shown as a quantitative indicator, such as elastography by ultrasound. Classification based on the impression of the surgeon is not sufficient, but rather the correlation with the pathological collagen fibrous content should be the main outcome in this study.
- Somatotroph adenomas should divide into densely granulated adenomas and sparsely granulated adenomas.
- Knosp grade and/or hardy classification is required in Table 2.
- It would be good to provide data on whether quantitative pharmacokinetic analysis of DCE-MRI is associated with prognosis such as residual tumor or recurrence.
Author Response
Comments;
- Definition of hard and soft of pituitary adenomas is ambiguous. If hardness is the main outcome, it should be shown as a quantitative indicator, such as elastography by ultrasound. Classification based on the impression of the surgeon is not sufficient, but rather the correlation with the pathological collagen fibrous content should be the main outcome in this study.
Thank you for your critical comment.
Unfortunately, elastography is not performed to evaluate pituitary tumor consistency in our institute. As mentioned in Discussion, qualitative assessment of tumor consistency is potential limitation of this study. Nonetheless, similar qualitative intraoperative assessments of pituitary tumor consistency have been used in many previous studies (Alimohamadi M.; et al. Acta Neurochir. 2014, 156, 2245–2252.; Pierallini A et al. Radiology. 2006, 239, 223–231.; Romano A et al. Pituitary. 2017, 20, 201–209.; Yiping L et al. J Neuroradiol. 2016, 43, 186–194.; Ma Z et al. Neuroradiology. 2016, 58, 51–57.; Iuchi T et al. Acta Neurochir. 1998, 140, 779–786.; Suzuki C et al. J Neuroradiol. 2007, 34, 228–235.; Mahmoud O.M et al. Eur J Radiol. 2011, 80, 412–417.).
We have added a brief discussion regarding this limitation:
Finally, assessment of tumor consistency was carried out qualitatively by a single operating surgeon, so possible observer bias cannot be excluded. Nonetheless, similar qualitative intraoperative assessments of pituitary tumor consistency have been used in many previous studies [4–11]. Moreover, the relationship between ve and tumor consistency was supported by the significant positive correlation between ve and histological collagen content.
- Somatotroph adenomas should divide into densely granulated adenomas and sparsely granulated adenomas.
Thank you for your critical comment.
All densely granulated adenomas were soft adenomas and all sparsely granulated adenomas were hard adenomas. We added the subtypes of GH producing adenomas in Table 2.
- Knosp grade and/or hardy classification is required in Table 2.
Thank you for your critical comment.
We have added the Knosp grading system in Materials and Methods chapter 2.5. Parasellar Extension on MRI (Grading System) and Knosp classification in Table 2.
2.5. Parasellar Extension on MRI (Grading System)
To evalutate radiological characteristics of cavernous sinus involvement, we used a grading system proposed by Knosp et al [28]. This grading system classifies the parasellar extension of pituitary adenomas on coronal MRI including precontrast T1- and T2-weighted images and postcontrast T1-weighted images. Three lines connecting the cross-section of the intracavernous and supracavernous internal carotid arteries distinguish 4 grades of parasellar adenoma extension: a medial tangent, a line through the cross-sectional centers, and a lateral tangent. The adenomas were divided into two groups according to the Knosp grading system: low grade with Knosp grade 0, 1 and 2 tumors and high grade with Knosp grade 3 and 4 tumors, because it was reported that the Knosp grade 0, 1 and 2 tumors demonstrated low rate (0%, 0% and 9.9%) and grade 3 and 4 tumors demonstrated high rate (37.9% and 100%) of cavernous sinus invasion [29].
- It would be good to provide data on whether quantitative pharmacokinetic analysis of DCE-MRI is associated with prognosis such as residual tumor or recurrence.
Thank you for your critical comment.
We have added a chapter for the postoperative MRI examinations in Materials and Methods (chapter 2.9). We have also added number of participants with residual tumor and tumor regrowth in Table 2.
2.9. Postoperative MRI examinations
Each participant in this study received follow-up MRI examinations 6 months postoperatively to determine extent of tumor resection (presence/absence of residual tumor), and yearly follow-up was performed thereafter. Tumor regrowth was positive when a tumor maximum diameter increased by more than 2 mm on MRI from the first follow-up MRI.
We have added the results of comparison of MRI parameters and PCC between the total resection and residual tumor and added new Table 5.
3.8. Comparisons of Imaging and Histologic Parameters between Total Resection and Residual Tumor
Mean values of the MRI parameters and PCC of the histological examination for total resection and residual tumor are shown in Table 5. There was no significant difference in any parameter.
Tumor regrowth was seen in only 4 cases, therefore we could not statistically compare MRI parameters and PCC between tumors with and without regrowth.
We have added following sentences in Discussion.
No imaging parameter was significantly different between totally resected tumors and those with residual tumor, and we failed to show that ve is a direct predictor for residual tumor. However, the mean value of ve of adenomas with residual tumor was higher than that of totally resected tumors. Therefore, it would be worth further studies as a candidate for predictor of surgical outcomes. Tumor regrowth was seen in only 4 cases, therefore we could not statistically compare MRI parameters and PCC between tumors with and without regrowth. Future studies with larger number of participants are required to clarify the roles of MRI parameters in predicting outcomes of pituitary adenoma.
Reviewer 2 Report
The authors aimed to study as to whether the preoperative mapping of volume of extravascular extracellular space per unit volume of tissue derived from pharmacokinetic dynamic contrast-enhanced MRI could be of any use for predicting the consistency of pituitary adenomas. The paper is in general interesting, and well-written, but there are some concerns that should be addressed by the authors:
- The authors mentioned that the adenoma max diameter ranged from 6mm to 69mm. How did the scan perform in such microadenomas? Is this reliable in such tiny adenomas? Perhaps the authors could should a comparison between micro vs macroadenomas (and/or giant adenomas, i.e. tumors larger than 4cm) to further illustrate its usefulness even in small adenomas, if that’s the case.
- The authors failed to show the difference in terms of MRI and PPC among the different pituitary tumor subtypes. Were there any differences between NFPAs and GHomas - for ACTHomas and TSHomas will be more difficult to assess because the number of samples is just too little, but at least these data should be shown for NFPAs and GHomas.
- Were any GHomas, or TSHomas, or the ACTHoma, treated with somatostatin analogues prior to the operation? This needs to be clearly stated in the manuscript, as somatostatin analogues have the potential to alter the biology of the tumor microenvironment and stromal tissue/fibroblasts eventually present within pituitary adenomas – for example by altering not only the hormonal secretion, but also potentially the cytokine and growth factor secretome (PMID: 31645017) which may ultimately influence the microenvironment and the composition of the extracellular matrix.
- The histopathology section is generally poorly described, and should be explained in more detail. How was the diagnosis of each pituitary tumor done? Were performed some IHC for pituitary transcription factors to define the lineage of the pituitary tumor, in line with the new classification of WHO from 2017 – as cited by these authors. What do the authors mean by “for clinically nonfunctioning adenoma, we did not evaluate immunohistochemistry”??? Were these tumors not stained to LH or FSH at least? And what about the pituitary transcription factors? How many of these were silent corticotroph adenomas or for instance null cell adenomas? Were there any differences among NFPAs regarding MRI and PPC features? These data should be added to the manuscript, even if only as a supplemental table(s). Was Ki67 staining done routinely, and if so did it correlate in any way with the MRI data or with the collagen content?
- In somatotrophinomas, were there differences between sparsely and densely granulated tumors? Was this actually searched at all? It would be important adding this because it has been suggested that sparsely-granulated GHomas are more fibrotic and difficult to remove during surgery.
- Were there any correlation between PCC and invasion of the cavernous sinus? Is the Azan staining sufficient to reliable tell about the consistency of a pituitary tumor, or would be appropriate to have some additional stainings, for a specific collagen, mesenchymal or fibroblast markers (like vimentin, FAP, FSP-1, aSMA)? A complementary histological study for further characterization of the extracellular matrix of this cohort would greatly enhance the relevance of the findings here reported.
- It is also not clear if all the 49 cases were operated for the first time? Or were any of these patients undergoing a second surgery? Any of the patients had before any sort of radiotherapy or other kinds of treatment? Did any patient had pituitary apoplexy?
- Were there any correlation between the levels of GH or IGF1 and the MRI data and PCC in the subgroup of somatotrophinomas? Could the authors add the biochemical data to the paper? This is relevant because GH is well known as an inducer of the epithelial-to-mesenchymal transition and may lead theoretically to more fibrotic tumors if this pathway is activated (see PMID: 28938477). Also surprising is that thyrotrophinomas are normally known by their fibrotic features (see PMID: 27125916), which was not the case in this series. Any explanation for this? Were these thyrotrophinoma patients treated pre-operatively with somatostatin analogues?
- Also, how many of the NFPA patients had hypopituitarism and which hormone deficits were present? This is also relevant data because the lack of hormones might influence the collagen composition of the tissue and also presumably the dynamics of the MRI (a patient with low cortisol may have decreased blood pressure and circulation which might be affecting the dynamic MRI parameters).
- The discussion could be improved, perhaps touching upon some of the elements raised above. What is the role of the pre-operative somatostatin analogues in those cases who were pre-treated before operation? What could be the role of the hormone excess in the MRI parameters and if could influence the consistency of the tumors itself? Among NFPAs, are there differences between the different histological subtypes?
- In lines 230-231: there are some numbers in yellow. Why? Also the volumes indicated for pituitary adenomas in general seems not to correlate with the data shown in Table 2.
- Mann Whitney U tests were used, but were the variables tested for normal distribution to decide whether parametric or non-parametric tests were used. This should be indicated in the statistical analysis section in methods
- The legends to Figure 2 and 3 should be revised, particularly the explanation of the immunohistochemical slide referred as to I).
- The asterisk added to the p value in the Tables to indicate Mann Whitney test was performed is confusing and somewhat misleading, eventually leading the readers to wrongly think that there are statistically significant differences. I would suggest this to be revised to enhance clarity.
Author Response
- The authors mentioned that the adenoma max diameter ranged from 6mm to 69mm. How did the scan perform in such microadenomas? Is this reliable in such tiny adenomas? Perhaps the authors could should a comparison between micro vs macroadenomas (and/or giant adenomas, i.e. tumors larger than 4cm) to further illustrate its usefulness even in small adenomas, if that’s the case.
Thank you for your critical comment.
For DCE-MRI, we used a prototype compressed sensing volumetric interpolated breath-hold examination sequence, which allowed for 0.9 x 0.9 x 0.9 mm3 high spatial resolution image.
We have only 3 microadenomas and 3 giant adenomas. Due to the small number of micro- and giant adenomas, statistical analyses were not performed.
We have changed a sentence in lines 227-229 as follows:
Three adenomas (6%) were smaller than 10 mm in maximum diameter (microadenoma), three adenomas (6%) were larger than 40 mm in maximum diameter (giant adenoma), while the remaining 43 (88%) were 10 mm or lager and 40 mm or smaller (macroadenoma).
We have replaced Figure 2 with a case with microadenoma (maximum diameter, 6 mm) which had soft consistency.
- The authors failed to show the difference in terms of MRI and PPC among the different pituitary tumor subtypes. Were there any differences between NFPAs and GHomas - for ACTHomas and TSHomas will be more difficult to assess because the number of samples is just too little, but at least these data should be shown for NFPAs and GHomas.
Thank you for your critical comment.
We have added the comparison of MRI parameters and PCC between the nonfunctioning adenomas and GH producing adenomas in the new Table 3.
We have also added following sentences in Results, and Discussion:
3.4. Comparisons of Imaging and Histologic Parameters between Nonfunctioning and GH producing Adenomas
Mean values of the MRI parameters and PCC for nonfunctioning and GH producing adenomas are shown in Table 3. GH producing adenomas had significantly lower rT2 than nonfunctioning adenomas (1.359 ± 0.334 vs. 1.763 ± 0.377, p < 0.01). No significant difference was found in any other MRI parameter or PCC.
Our study showed that the rT2 of GH producing adenomas was significantly lower than that of nonfunctioning adenomas, which is compatible with literature [44,45]. It was previously reported that T2 signal intensity is correlated with granulation pattern, collagen content, degree of fibrosis and amyloid accumulation in GH producing adenomas [44,45].
- Were any GHomas, or TSHomas, or the ACTHoma, treated with somatostatin analogues prior to the operation? This needs to be clearly stated in the manuscript, as somatostatin analogues have the potential to alter the biology of the tumor microenvironment and stromal tissue/fibroblasts eventually present within pituitary adenomas – for example by altering not only the hormonal secretion, but also potentially the cytokine and growth factor secretome (PMID: 31645017) which may ultimately influence the microenvironment and the composition of the extracellular matrix.
Thank you for your critical comment.
One participant with TSH producing adenoma was treated with somatostatin analogues prior to the operation, and the tumor had soft consistency.
We have added this information in results chapter 3.1. Characteristics of Participants and Adenomas.
We have also added a brief discussion on this topic.
In our series, only one participant received preoperative treatment with somatostatin analogues, which may possibly influence microenvironment in adenoma by altering hormonal secretion, cytokine, and growth factor secretome [48]. A previous study reported that there was no significant change in T2 signal intensity ratio between before and after somatostatin analogue treatment in GH producing adenomas [49]. Effects of somatostatin analogue treatment on MRI parameters and tumor consistency need to be clarified in future studies.
- The histopathology section is generally poorly described, and should be explained in more detail. How was the diagnosis of each pituitary tumor done? Were performed some IHC for pituitary transcription factors to define the lineage of the pituitary tumor, in line with the new classification of WHO from 2017 – as cited by these authors. What do the authors mean by “for clinically nonfunctioning adenoma, we did not evaluate immunohistochemistry”??? Were these tumors not stained to LH or FSH at least? And what about the pituitary transcription factors? How many of these were silent corticotroph adenomas or for instance null cell adenomas? Were there any differences among NFPAs regarding MRI and PPC features? These data should be added to the manuscript, even if only as a supplemental table(s). Was Ki67 staining done routinely, and if so did it correlate in any way with the MRI data or with the collagen content?
Thank you for your critical comment.
In WHO 2017 classification, all pituitary adenomas should be evaluated using immunohistochemistry; however, we did not evaluate immunohistochemistry in nonfunctioning adenomas. We did not evaluate the pituitary transcription factors or Ki67, either. Instead, in this study, each participant was diagnosed according to the guideline of diagnosis and treatment of hypothalamic pituitary dysfunction published by The Japan Endocrine Society in 2019. Each adenoma was classified into either non-functioning, GH producing, TSH producing, or ACTH producing based on endocrine studies.
We have deleted the description about WHO 2017 guideline in Histologic Examination, and added sentences in 2.1. Participant Characteristics as follow:
The diagnoses of all the participants were confirmed on the hormonal activity on the bases of guidelines of diagnosis and treatment of hypothalamic pituitary dysfunction [23].
Accordingly, we have changed the diagnoses of functioning adenomas in the text and Tables from somatotroph, thyrotroph, and corticotroph into GH producing, TSH producing, and ACTH producing, respectively.
In addition, we added some more details of histopathological examination in Materials and Methods chapter 2.8. Histologic Examinations as follows:
Tissues were fixed in 4% paraformaldehyde in 0.05 M phosphate buffer, pH 7.4, followed by immersion in 30 % sucrose in 0.05 M phosphate buffer, pH 7.2. Tissues were then processed into cryoblocks with liquid nitrogen and kept in deep freeze. A cryostat was used to obtain 8 μm cryosections, and the tissues were then mounted on glass slides. Tissue conditions were confirmed by hematoxylin–eosin. Azan staining was performed to detect fibrous matrix deposition (Azan staining displays fibrous matrix in blue; other tissue regions are stained in red or purple).
We have also added following sentences in Discussion:
We did not perform immunohistochemistry analyses for pituitary transcription factors to define the lineage of the pituitary tumors according to WHO 2017 classification. Instead, each participant was diagnosed according to the guideline of diagnosis and treatment of hypothalamic pituitary dysfunction published by The Japan Endocrine Society in 2019 [23]. Correlations between the full classification of the pituitary adenoma and MRI parameters remain to be a subject of future studies.
- In somatotrophinomas, were there differences between sparsely and densely granulated tumors? Was this actually searched at all? It would be important adding this because it has been suggested that sparsely-granulated GHomas are more fibrotic and difficult to remove during surgery.
Thank you for your critical comment.
All densely granulated adenomas were soft adenomas and all sparsely granulated adenomas were hard adenomas. We have added the GH producing adenoma subtype in Table 2.
- Were there any correlation between PCC and invasion of the cavernous sinus? Is the Azan staining sufficient to reliable tell about the consistency of a pituitary tumor, or would be appropriate to have some additional stainings, for a specific collagen, mesenchymal or fibroblast markers (like vimentin, FAP, FSP-1, aSMA)? A complementary histological study for further characterization of the extracellular matrix of this cohort would greatly enhance the relevance of the findings here reported.
Thank you for your critical comment.
We have added the Knosp grading system in Materials and Methods chapter 2.5. Parasellar Extension on MRI (Grading System) and Knosp classification in Table 2.
We have added the results of comparison of MRI parameters and PCC between tumor with low and high grade Knosp classification in the new Table 4.
3.7. Comparisons of Imaging and Histologic Parameters between adenomas with Low and High Grade of Knosp classification
Mean values of the MRI parameters and PCC of the histological examination for the tumor with low and high grade of Knosp classification are shown in Table 4. High grade of Knosp classification adenomas had significantly higher PCC than low grade of Knosp classification adenomas (25.4 ± 22.3 vs. 12.6 ± 15.3, p = 0.03). No significant difference was found in any other MRI parameter.
Several authors (Naganuma et al. Neurol Med Chir. 2002, 42, 202–212, Tofrizal A et al. Med Mol Morphol. 2016, 49, 224–232. Tofrizal A et al. Med Mol Morphol. 2017 , 50, 145–154.) used Azan staining to reveal that fibrous connective tissues in firm pituitary adenomas mainly consisted of collagen. Therefore, we assumed that Azan staining is a feasible method to evaluate collagen content of pituitary adenomas.
We added a sentence in Discussion as follows:
On the basis of Azan, van Gieson, and Sirius red staining methods, previous studies have reported mean percentages of collagen contents of hard adenomas were significantly higher than those of soft adenomas [9,33–35].
We did not perform staining for a specific collagen, mesenchymal or fibroblast markers in this study.
- It is also not clear if all the 49 cases were operated for the first time? Or were any of these patients undergoing a second surgery? Any of the patients had before any sort of radiotherapy or other kinds of treatment? Did any patient had pituitary apoplexy?
Thank you for your critical comment.
We have added following sentences in Results (3.1. Characteristics of Participants and Adenomas):
Three participants had recurrent tumor and were undergoing a second surgery, and all three tumors had hard consistency. No other participants had any prior treatment including radiotherapy. No participants had pituitary apoplexy.
- Were there any correlation between the levels of GH or IGF1 and the MRI data and PCC in the subgroup of somatotrophinomas? Could the authors add the biochemical data to the paper? This is relevant because GH is well known as an inducer of the epithelial-to-mesenchymal transition and may lead theoretically to more fibrotic tumors if this pathway is activated (see PMID: 28938477). Also surprising is that thyrotrophinomas are normally known by their fibrotic features (see PMID: 27125916), which was not the case in this series. Any explanation for this? Were these thyrotrophinoma patients treated pre-operatively with somatostatin analogues?
Thank you for your critical comment.
We have added the methods of endocrinological evaluation in Materials and Methods.
2.2. Endocrine Studies
Blood basal levels of the anterior pituitary hormones (growth hormone [GH], thyroid stimulating hormone [TSH], prolactin [PRL], lute-inizing hormone [LH], follicle-stimulating hormone [FSH], and adrenocorticotropic hormone [ACTH]) and their target hormones (insu-lin-like growth factor 1 [IGF-1], free thyroxine [FT4], free triiodothyronine [FT3], testosterone, and estradiol) were measured pre- and postoperatively. Pituitary stimulation tests were also performed in some cases pre- and postoperatively using a combination of thy-rotropin-releasing hormone (TRH) (500 μg), LH-releasing hormone (100 μg), and corticotropin-releasing hormone (100 μg). A chemiluminescent enzyme immunoassay was used to measure TSH (normal range 0.50–5.00 μIU/ml). The tumor was considered to be TRH responsive if the TSH level increased more than twice the basal level in response to the TRH stimulation test. Cosecretion of GH was identified by supranormal IGF-1 levels and a lack of GH suppression in response to a 75-g glucose tolerance test. Associated hyperprolactinemia was identified when tumor cells showed prolactin immunopositivity. GH hypersecretion was considered to be in complete remission when fulfilling the conditions of a normal basal GH level, normal GH suppression (GH nadir < 0.4 ng/ml) during glucose tolerance testing, and normal IGF-1 levels based on age and sex [24]. Octreotide (50 μg administered subcutaneously) or bromocrip-tine (2.5 mg by mouth) tests were performed to investigate TSH responses to somatostatin analogs or dopamine agonists. TSH was considered suppressed if it decreased to less than 50% of the basal level. Plasma TSH, FT3, and FT4 levels were measured 2–4 times during the 2-week hospital stay after surgery.
We have added the correlation between the level of GH and the MRI data and PCC in the subgroup of GH producing adenomas.
3.5. Correlation of Imaging and Histologic Parameters, and Levels of GH and IGF-1 in Participants with GH producing Adenoma
We found a significant positive correlation between vp and level of GH in GH producing adenomas (n = 13, r = 0.630, p = 0.021). No other MRI parameters or PCC showed a significant correlation with GH (rT1 [r = 0.402, p = 0.17], rT2 [r = 0.123, p = 0.69], ADC [r = −0.256, p = 0.40], ve [r = −0.480, p = 0.10], Ktrans [r = −0.424, p = 0.15], kep [r = −0.333, p = 0.27], and PCC [r = −0.273, p = 0.37]) or IGF-1 (rT1 [r = 0.294, p = 0.33], rT2 [r = −0.019, p = 0.95], ADC [r = −0.299, p = 0.32], ve [r = −0.497, p = 0.08], vp [r = 0.391, p = 0.19], Ktrans [r = −0.397, p = 0.18], kep [r = −0.149, p = 0.63], and PCC [r = −0.308, p = 0.31]).
We have added a brief discussion regarding these results as follows:
Previous studies have implicated GH in the progression of several cancers, including breast, colorectal, and pancreatic [39-41]. A mechanism by which GH may play this role in cancer is through the induction of the epithelial-to-mesenchymal transition [42] and may lead theoretically to increase in fibroblasts [42,43]. However, our study failed to show a significant correlation between the GH level and PCC. The GH level showed a significant positive correlation with vp, which is blood plasma volume per unit volume of tissue. The underlying mechanism of this correlation is unknown.
One participant with TSH producing adenoma was treated pre-operatively with somatostatin analogue. We have added a brief discussion regarding TSH producing adenomas as follows:
There were two TSH producing adenomas in our series, and both had soft consistency. It was previously reported that TSH producing adenomas are characterized by fibrosis and hard tumor consistency [46,47], which was not the case in our series. Future studies with larger number of participants are required to clarify this point.
- Also, how many of the NFPA patients had hypopituitarism and which hormone deficits were present? This is also relevant data because the lack of hormones might influence the collagen composition of the tissue and also presumably the dynamics of the MRI (a patient with low cortisol may have decreased blood pressure and circulation which might be affecting the dynamic MRI parameters).
Thank you for your critical comment.
We have added the following sentences in Results.
3.6. Hypopituitarism and Correlation between Cortisol Level and Imaging and Histologic Parameters in Nonfunctioning Adenomas
Twenty-one (63.3%) participants with nonfunctioning adenoma had hypopituitarism: 11 (52.4%) participants had cortisol deficits, 3 (14.3%) had FSH deficits, 3 (14.3%) had undergone thyroid agent treatment, and 4 (19.0%) had diabetes insipidus. No MRI parameter or PCC showed a significant correlation with cortisol (rT1 [r = 0.055, p = 0.76], rT2 [r = −0.069, p = 0.70], ADC [r = −0.036, p = 0.84], ve [r = −0.052, p = 0.77], vp [r = 0.136, p = 0.45], Ktrans [r = −0.059, p = 0.75], kep [r = −0.073, p = 0.69], and PCC [r = −0.118, p = 0.51]).
Correspondingly, we have also added the following sentences in Discussion.
In our study, the cortisol level did not influence MRI parameters or PCC in participants with nonfunctioning adenomas. This result did not support a hypothesis that participants with low cortisol may have decreased blood pressure and circulation which might be affecting the dynamic MRI parameters.
Also, we have added the numbers of participants with hypopituitarism in the groups of soft adenoma and hard adenoma in Table 2. Hypopituitarism was significantly more frequent in the group of hard adenoma. We have added this result in the text (3.1. Characteristics of Participants and Adenomas).
- The discussion could be improved, perhaps touching upon some of the elements raised above. What is the role of the pre-operative somatostatin analogues in those cases who were pre-treated before operation? What could be the role of the hormone excess in the MRI parameters and if could influence the consistency of the tumors itself? Among NFPAs, are there differences between the different histological subtypes?
Thank you for your critical comment.
As mentioned above, we have added the discussion regarding i) roles of preoperative somatostatin analogues in the MRI parameters and PCC and ii) roles of hormone excess in the MRI parameters and PCC. As for the differences in the MRI parameters and PCC between different histological subtypes of NFPAs, we did not evaluate immunohistochemistry of hormones or pituitary transcription factors for NFPAs.
i) Roles of preoperative somatostatin analogues in the MRI parameters and PCC
In our series, only one participant received preoperative treatment with somatostatin analogue, which may possibly influence microenvironment in adenoma by altering hormonal secretion, cytokine, and growth factor secretome [48]. A previous study reported that there was no significant change in T2 signal intensity ratio between before and after somatostatin analogue treatment in GH producing adenomas [49]. Effects of somatostatin analogue treatment on MRI parameters and tumor consistency need to be clarified in future studies.
ii) Roles of hormone excess in the MRI parameters and PCC
Previous studies have implicated GH in the progression of several cancers, including breast, colorectal, and pancreatic [39-41]. A mechanism by which GH may play this role in cancer is through the induction of the epithelial-to-mesenchymal transition [42] and may lead theoretically to increase in fibroblasts [42,43]. However, our study failed to show a significant correlation between the GH level and PCC. The GH level showed a significant positive correlation with vp, which is blood plasma volume per unit volume of tissue. The underlying mechanism of this correlation is unknown.
- In lines 230-231: there are some numbers in yellow. Why? Also the volumes indicated for pituitary adenomas in general seems not to correlate with the data shown in Table 2.
I could not find any yellow numbers in my manuscript. I hope that the numbers appear normally in the revised manuscript.
I believe the volume data in the text are exactly the same as those in Table 2.
- Mann–Whitney U tests were used, but were the variables tested for normal distribution to decide whether parametric or non-parametric tests were used. This should be indicated in the statistical analysis section in methods
Thank you for your critical comment.
We have corrected the statistical analysis as follows:
Relationships between categorical variables were tested using either the Chi–square test or Fisher’s exact test. Comparisons between numerical variables were preformed using either the Mann–Whitney U test or unpaired t–test. We used the D'Agostino–Pearson normality test to check the normality of the data. Pearson’s correlation coefficients were used to analyze correlations between the MRI parameters, PCC, and hormone levels.
We have also changed p values in Table 6.
- The legends to Figure 2 and 3 should be revised, particularly the explanation of the immunohistochemical slide referred as to I).
Thank you for your critical comment.
We have modified legends for the immunohistochemical slides in Figures 2 and 3 as follows:
Figure 2
Azan-stained section of tumor at pathologic examination shows abundant collagen in the stromal fibrous tissue in blue and small tumor cells in red.
Figure 3
Azan-stained section of tumor at pathologic examination shows scant collagen in the stromal fibrous tissue in blue within predominant tumor cells in red.
- The asterisk added to the p value in the Tables to indicate Mann Whitney test was performed is confusing and somewhat misleading, eventually leading the readers to wrongly think that there are statistically significant differences. I would suggest this to be revised to enhance clarity.
Thank you for your critical comment.
We have changed the asterisks in Tables to alphabets.
Reviewer 3 Report
A significant limitation of complete pituitary tumor resection is partially related to consistency. To this end, the authors examine the ability of quantitative dynamic contrast enhanced MRI (DCE-MRI) to predict tumor consistency. They show that pharmacokinetic parameters and the histologic percentage of collagen content tent (PCC) can be correlated.
The study is of interest but can be significantly improved. They must focus on the true clinical relevance of their findings.
- Please provide full tumor classification including the non-functional ones. These are the tumors where surgery plays the biggest role and hence complete resection is most critical.
- Please identify a metric of completeness of resection. Hardness is a term of little significance unless it is corroborated by finding of relevance. Outcomes have to be measured by the extent to which pre-operative DCE-MRI is able to predict difficulty with resection. Otherwise, what is the point?
- Another outcome of potential interest will be the extent to which DCE-MRI can predict tumor re-growth.
- How does PCC relate to other measures of pituitary tumor behaviour or invasiveness?
Author Response
- Please provide full tumor classification including the non-functional ones. These are the tumors where surgery plays the biggest role and hence complete resection is most critical.
Thank you for your critical comment.
Unfortunately, we did not evaluate immunohistochemistry of hormones or pituitary transcription factors for non-functioning pituitary adenomas. We have added a brief discussion regarding this limitation in Discussion:
We did not perform immunohistochemistry analyses for pituitary transcription factors to define the lineage of the pituitary tumors according to WHO 2017 classification. Instead, each participant was diagnosed according to the guideline of diagnosis and treatment of hypothalamic pituitary dysfunction published by The Japan Endocrine Society in 2019 [23]. Correlations between the full classification of the pituitary adenoma and MRI parameters remain to be a subject of future studies.
- Please identify a metric of completeness of resection. Hardness is a term of little significance unless it is corroborated by finding of relevance. Outcomes have to be measured by the extent to which pre-operative DCE-MRI is able to predict difficulty with resection. Otherwise, what is the point?
- Another outcome of potential interest will be the extent to which DCE-MRI can predict tumor re-growth.
Thank you for your critical comments.
We have added a chapter for the postoperative MRI examinations in Materials and Methods (chapter 2.9). We have also added number of participants with residual tumor and tumor regrowth in Table 2.
2.9. Postoperative MRI Examinations
Each participant in this study received follow-up MRI examinations 6 months postoperatively to determine extent of tumor resection (presence/absence of residual tumor), and yearly follow-up was performed thereafter. Tumor regrowth was positive when a tumor maximum diameter increased by more than 2 mm on MRI from the first follow-up MRI.
We added the results of comparison of MRI parameters and PCC between the total resection and residual tumor and added new Table 5.
3.8. Comparisons of Imaging and Histologic Parameters between Total Resection and Residual Tumor
Mean values of the MRI parameters and PCC of the histological examination for total resection and residual tumor are shown in Table 5. There was no significant difference in any parameter.
Tumor regrowth was seen in only 4 cases, therefore we could not statistically compare MRI parameters and PCC between tumors with and without regrowth.
We have added following sentences in Discussion.
No imaging parameter was significantly different between totally resected tumors and those with residual tumor, and we failed to show that ve is a direct predictor for residual tumor. However, the mean value of ve of adenomas with residual tumor was higher than that of totally resected tumors. Therefore, it would be worth further studies as a candidate for predictor of surgical outcomes. Tumor regrowth was seen in only 4 cases, therefore we could not statistically compare MRI parameters and PCC between tumors with and without regrowth. Future studies with larger number of participants are required to clarify the roles of MRI parameters in predicting outcomes of pituitary adenoma.
- How does PCC relate to other measures of pituitary tumor behaviour or invasiveness?
Thank you for your comment.
We have added the Knosp grading system in Materials and Methods chapter 2.5. Parasellar Extension on MRI (Grading System) and Knosp classification in Table 2.
Also we have added the comparison of MRI parameters and PCC between tumor with low and high grade Knosp classification in the new Table 4.
3.7. Comparisons of Imaging and Histologic Parameters between the Low and High Grade of Knosp classification
Mean values of the MRI parameters and PCC of the histological examination for the low and high grade of Knosp classification are shown in Table 4. High grade of Knosp classification adenomas had significantly higher PCC than low grade of Knosp classification adenomas (25.4 ± 22.3 vs. 12.6 ± 15.3, p = 0.03). No significant difference was found in any other MRI parameter.
Reviewer 4 Report
Kamimura et al. prospectively investigated whether DCE-MRI is useful tool for predicting consistency of pituitary adenoma. This is a properly written paper; however, some issues need to be clarified before the paper could be considered for publication. 1. The criteria for dividing soft and hard tissue are ambiguous. Please explain whether it was divided by objective standard or whether the surgeon's subjective standard worked. Without objective criteria for the classification of these two, the current analysis has a major limitation as the authors also mentioned as a limitation. 2. There was no significant difference in the characteristics of adenoma between the soft group and the hard group, but more functional adenomas were included in the soft group. I am curious about the authors' opinions on the difference in analysis according to the nature of adenoma. I am also curious about the opinions of the authors on separately classifying and analyzing NFPA only. 3. In table 2, please rewrite the percentage uniformly by row or column.Author Response
- The criteria for dividing soft and hard tissue are ambiguous. Please explain whether it was divided by objective standard or whether the surgeon's subjective standard worked. Without objective criteria for the classification of these two, the current analysis has a major limitation as the authors also mentioned as a limitation.
Thank you for your comment.
Unfortunately, we did not use any quantitative technique for measurement of consistency such as elastography. The tumors were classified into groups of soft and hard consistency: tumors with soft were easily removable through suction, and those with hard consistency were removable with difficulty through suction or not removable through suction but excisable piece by piece As mentioned in Discussion, qualitative assessment of tumor consistency is potential limitation of this study, although similar qualitative intraoperative assessments of pituitary tumor consistency have been used in many previous studies (Alimohamadi M.; et al. Acta Neurochir. 2014, 156, 2245–2252.; Pierallini A et al. Radiology. 2006, 239, 223–231.; Romano A et al. Pituitary. 2017, 20, 201–209.; Yiping L et al. J Neuroradiol. 2016, 43, 186–194.; Ma Z et al. Neuroradiology. 2016, 58, 51–57.; Iuchi T et al. Acta Neurochir. 1998, 140, 779–786.; Suzuki C et al. J Neuroradiol. 2007, 34, 228–235.; Mahmoud O.M et al. Eur J Radiol. 2011, 80, 412–417.).
We have added a brief discussion regarding this limitation:
Finally, assessment of tumor consistency was carried out qualitatively by a single operating surgeon, so possible observer bias cannot be excluded. Nonetheless, similar qualitative intraoperative assessments of pituitary tumor consistency have been used in many previous studies [4–11]. Moreover, the relationship between ve and tumor consistency was supported by the significant positive correlation between ve and histological collagen content.
- There was no significant difference in the characteristics of adenoma between the soft group and the hard group, but more functional adenomas were included in the soft group. I am curious about the authors' opinions on the difference in analysis according to the nature of adenoma. I am also curious about the opinions of the authors on separately classifying and analyzing NFPA only.
Thank you for your constructive comments.
We have added comparisons MRI parameters and PCC between soft and hard nonfunctioning adenoma and in Results:
3.10. Comparisons of Imaging and Histologic Parameters between Soft and Hard Nonfunctioning Adenomas
Mean values of the MRI parameters and PCC of the histological examination for soft and hard nonfunctioning adenomas are shown in Table 7. Hard nonfunctioning adenomas had significantly higher PCC (43.08 ± 15.91 vs. 6.62 ± 3.51, p < 0.01) and ve (0.310 ± 0.114 vs. 0.215 ± 0.118, p = 0.03), and significantly lower vp (0.036 ± 0.043 vs. 0.089 ± 0.096, p = 0.04) and rT2 (1.558 ± 0.273 vs. 1.880 ± 0.383, p = 0.01) values than soft nonfunctioning adenomas, whereas there were no significant differences in the other MRI parameters (Table 7).
3.11. MRI Parameters Correlated with Percentage of Collagen Content in Pituitary Adenomas
We found a significant positive correlation between ve and PCC in pituitary adenomas (n = 49, r = 0.601, p < 0.01; Figure 4). No other MRI parameters showed a significant correlation with PCC (rT1 [r = 0.035, p = 0.81], rT2 [r = −0.075, p = 0.61], ADC [r = 0.044, p = 0.76], vp [r = −0.165, p = 0.26], Ktrans [r = 0.057, p = 0.70], and kep [r = −0.055, p = 0.71]).
In addition, we found a significant positive correlation between ve and PCC in nonfunctioning adenomas (n = 33, r = 0.567, p < 0.01), whereas no significant correlation with PCC was found in the other MRI parameters (rT1 [r = 0.157, p = 0.38], rT2 [r = −0.309, p = 0.08], ADC [r = −0.063, p = 0.73], vp [r = −0.238, p = 0.18], Ktrans [r = 0.033, p = 0.85], and kep [r = −0.045, p = 0.81]).
In addition, we have added following discussion regarding these results:
In our study, none of rT1, rT2, or ADC was significantly correlated with tumor consistency or collagen content, when all adenoma subtypes were included in the analysis. However, the present study revealed that rT2 of hard nonfunctioning adenoma was significantly lower than that of soft nonfunctioning adenoma, while no significant difference was noted in rT1 nor ADC. It was suggested that, in addition to ve, rT2 may be useful in predicting consistency of nonfunctioning adenomas.
- In table 2, please rewrite the percentage uniformly by row or column.
Thank you for your comment.
We have rewritten the percentage uniformly by column.